# Validation of the Spanish-Mexican Version of the Australian Breastfeeding Attitude Questionnaire in Higher Education Health Students

**DOI:** 10.3390/ijerph18094609

**Published:** 2021-04-27

**Authors:** Gabriela Alejandra Grover-Baltazar, Gabriela Macedo-Ojeda, Ana Sandoval-Rodríguez, Marianne Martínez-Vizmanos, Lucrecia Carrera-Quintanar, Barbara Vizmanos

**Affiliations:** 1Cuerpo Académico UDG-CA-454 “Alimentación y Salud en el proceso Salud-Enfermedad”, Departamento de Clínicas de la Reproducción Humana, Crecimiento y Desarrollo Infantil, Centro Universitario de Ciencias de la Salud (CUCS) Universidad de Guadalajara (UdeG), México, Calle Hospital 320, Colonia El Retiro, Guadalajara 44280, Jalisco, Mexico; grover_1905@hotmail.com (G.A.G.-B.); gabriela.macedo@cucs.udg.mx (G.M.-O.); anasol44@hotmail.com (A.S.-R.); mariana.martinez3284@alumnos.udg.mx (M.M.-V.); lucrecia.carrera@gmail.com (L.C.-Q.); 2Instituto de Investigación en Ciencias Biomédicas (IICB), CUCS, UdeG, Sierra Mojada 950, edificio Q, Colonia Independencia, Guadalajara 44340, Jalisco, Mexico; 3Instituto de Biología Molecular en Medicina, CUCS, UdeG, Sierra Mojada 950, edificio Q, Colonia Independencia, Z.C., Guadalajara 44340, Jalisco, Mexico; 4Instituto de Nutrigenética y Nutrigenómica Traslacional, CUCS, UdeG, Sierra Mojada 950, edificio Q, Colonia Independencia, Guadalajara 44340, Jalisco, Mexico

**Keywords:** breastfeeding, breastfeeding attitudes questionnaire, questionnaire validation, validation study, health students, Spanish language

## Abstract

Positive attitudes towards breastfeeding in health professionals/students have been associated with increasing their confidence to provide support and accompaniment to mothers. In Mexico, there is no valid/reliable tool to assess attitudes towards breastfeeding in this population. The Australian Breastfeeding Attitudes (and Knowledge) Questionnaire (ABAQ) measures attitudes in the Australian population. We aimed to adapt and validate the ABAQ in Mexican health students. We included 264 health students (nursing, nutrition, and medicine) from the University of Guadalajara. Bilingual translators carried out the Spanish adaptation with a reverse translation into English. Experts evaluated the content validity. Reliability was evaluated through an internal consistency analysis (Cronbach’s alpha) and construct validity through convergent–divergent validation, item–total correlation, exploratory factor analysis (by principal components), and confirmatory factor analysis. According to the exploratory factor analysis, only one component was identified. Seven items were removed (low correlation between items ≤0.2 and low factor load ≤0.3). The Cronbach’s alpha was 0.78. According to the confirmatory factor analysis, the one-factor solution of the ABAQ-13Mx showed a good model fit (*X*^2^ = 98.41, G = 62, *p* = 0.02, CFI = 0.940, and RMSEA = 0.048). The ABAQ-13Mx is a reliable and valid instrument for evaluating attitudes towards breastfeeding in Mexican health degree students.

## 1. Introduction

Breastfeeding (BF) is the gold standard for newborn and infant feeding [1]. There is evidence of short- and long-term health benefits of BF for infants and mothers [2]. In children, BF reduces morbidity and mortality due to infectious diseases, decreases the prevalence of diabetes and obesity [3,4], and in women, it is associated with a lower risk of breast and ovarian cancers and diabetes [3], to mention but a few of the benefits. 

According to the Mexican National Health and Nutrition Survey (ENSANUT), the prevalence of exclusive breastfeeding in 2018 was 28.6% in Mexico. Only 46.9% of children were estimated to have continuous BF until their first year of life and 29% until two years of age [5]. Although these data improved compared to the 2012 survey, the values are far from those proposed by the WHO for 2030 [6]. Mexico is one Spanish-speaking country that needs to reinforce BF practices, but many others also require more studies and interventions to improve BF occurrences.

Health professionals, including students, play a crucial role in a mother’s decision to breastfeed or not. In consequence, they should provide evidence-based information on several areas of BF practice, in addition to accompanying them during and after delivery [7,8]. The attitudes and knowledge of health science students towards BF greatly influence the effectiveness of BF promotion [9]. Although knowledge is an indispensable element to support BF, attitudes play a pivotal role, since they are predictors of the student’s behavior; thus, the more positive the attitude, the more favorable the behavior [10,11,12].

Attitudes are predispositions towards a social object, which can be learned or innate. The individual will react in an evaluative, favorable, or unfavorable way to that social object [13]. Consequently, attitudes are internal constructions of the individual and are part of his affective component [14]. There are different instruments used to assess the attitudes toward BF in health students, most of them in English [15,16]. The former [16] has been validated in Spanish in different countries (Mexico, Colombia, and Spain, respectively) [17,18,19] but is centered on pregnant or breastfeeding women. The later paper [15], according to the literature review, the Australian Breastfeeding Knowledge and Attitude Questionnaire (ABKAQ) is one of the most widely used questionnaires to assess attitudes in health professionals/students [15]. This questionnaire has been validated in different languages, such as Chinese [20] and English, for different countries [9,12,15,21,22] but not in Spanish. 

The ABKAQ evaluates two components/dimensions: attitudes and knowledge (20 and 40 items, respectively), with a Likert-type scale of 1–5 points ABKAQ, in Australian general practitioners, has shown an adequate internal consistency (Cronbach’s alpha 0.84 and 0.83, respectively, in attitudes and knowledge). Content validity was achieved with a panel of four LM experts [15].

However, in Mexico, to our knowledge, there is no valid and reliable instrument to evaluate this construct in health students. The lack of instruments demonstrates the need to provide tools in Spanish to assess the attitudes toward BF to benefit more than 480 million Spanish-speaking inhabitants in the world through their health. This study aimed to adapt the attitude component of the Australian Breastfeeding (Knowledge) Attitude Questionnaire (ABAQ) to Spanish, evaluate its psychometric properties, and validate it in Mexican health students.

## 2. Materials and Methods

The present validation study used a cross-sectional quantitative methodology. The study was evaluated by the Research Ethics Committees and by the Research Committee of the University Center of Health Sciences of the University of Guadalajara (CUCS-UdeG) and considered international and national ethical guidelines in its execution. 

The translation and validation of the attitude component of the ABAQ were conducted in two phases.

### 2.1. Phase I. Translation and Cultural Adaptation of the ABAQ Questionnaire

This phase adopts the methodology proposed by Carvajal, Centeno, and Watson et al. (2011) [23]. First, the original English questionnaire was translated independently by two certified bilingual translators whose native language is Spanish. Both translations were compared to identify discrepancies; they were discussed until consensus was reached. Some words were modified or substituted to fit the context better. The order and structure of some sentences were agreed upon to facilitate reading and comprehension. After this consensus, the version was translated into English (back translation) by an expert translator. The author of the ABAQ test evaluated this version to confirm that it was conceptually equivalent to the original. The Spanish version was evaluated by five health professionals, experts with experience in BF and who teach or have taught in the three health programs mentioned. They evaluated the clarity and congruence of the items (each item was scored with 0 = no or 1 = yes). The score for each characteristic was summed for each item; the scores ranged from 0 to 5 points. Values <3 would be considered “unacceptable” (they should be modified and reevaluated), scores of 3 or 4 “acceptable”, and values of 5 as “optimal”.

After the expert’s analysis, the questionnaire was applied to a pilot group of health students (*n* = 11). They voluntarily completed the questionnaire and made comments per item, pointing out aspects they considered difficult to understand (words or the item’s sentence). Minimal changes were made to clarify the items’ expression, thus conforming to the version of the questionnaire to be validated.

### 2.2. Phase II. Evaluation of Psychometric Properties and Validation

According to the procedures established for psychometric tests [24], we evaluated the instrument’s reliability (internal consistency) with Cronbach’s alpha. Construct validity was assessed by convergent–divergent validity, exploratory factor analysis (EFA) through principal components analysis (PCA), and confirmatory factor analysis (CFA).

### 2.3. Description of the ABAQ-Mx Instrument

The questionnaire adapted to Spanish, ABAQ-Mx, evaluates the attitude component of the original questionnaire. It is designed to be self-administered (20 items) with Likert-type responses. It ranges from “strongly disagree” (1) to “strongly agree” (5). Of the 20 items, 15 were reverse-worded; thus, scores of “strongly disagree” (1) indicate a positive attitude toward BF.

### 2.4. Participants and Procedures 

Ten participants were considered for each item [24]; as there were 20 items, 200 students were considered adequate. The inclusion criteria were: to be a student (at least in the second semester of undergraduate studies related to the mother–baby–father triad) in the health area at CUCS-UdeG, Mexico. The University of Guadalajara is made up of a network of university centers distributed throughout Jalisco. It is a public university, a leader in the formation of human resources [25]. It was ranked in a world ranking as the third best university in Mexico [26]. CUCS-UdeG is part of this network and annually trains almost eleven thousand bachelor degree health students, most of them in medicine, nutrition, and nurse programs, corresponding to 67.6% of all undergraduate students, only in CUCS-UdeG [27]. For many years it has had the most significant number of research professors recognized nationally by the National System of Researchers. The CUCS UdeG is part of this state network, where health professionals are trained: medicine, nursing, and nutrition, among others. All three programs are on the list of high-performance programs of the national Mexican CENEVAL’s General Undergraduate Exit Examination, which is another external evaluation instance upon completion of the credits.

Randomly, at different times and buildings of the CUCS-UdeG, upon request to the professor, students of one of these three programs were invited to answer the questionnaire as part of their class time. They were verbally requested to participate, and it was emphasized that it was voluntary, anonymous, and did not affect their course grade. The volunteers received an informed consent document. Those who signed it, after clarification, received the printed survey to answer, with no pre-established time limit. The approximate response time was 15–20 min.

### 2.5. Data Analysis

Statistical analysis was performed using IBM-SPSS Statistics and IBM-Amos Graphics software (Version 22.0, Chicago, IL, USA, 2013). 

Reliability, by means of internal consistency (Cronbach’s alpha), considered the following criteria: questionable (0.60–0.69), acceptable (0.70–0.79), good (0.80–0.89), and redundant (0.90–1) [28]. The validation of the construct was evaluated in two stages. First, using the convergence–divergence method (Spearman’s correlation coefficient): an item–total correlation ≤0.2 was considered an elimination criterion [29]. The PCA (principal component analysis extraction method with varimax rotation) was then performed, considering factor loadings ≤0.3 as an elimination criterion [30]. In order to evaluate that the ABAQ-Mx scale data were adequate, Bartlett’s test of sphericity was used, considering a *p*-value ≤ 0.05 as acceptable and a Kaiser–Meyer–Olkin (KMO) measure of sampling adequacy as adequate, if it was ≥0.6 [30].

A confirmatory factor analysis (CFA) was performed to determine whether the factor structure was replicable for the Mexican population. The adequacy of the model was estimated, employing goodness-of-fit indices. We considered the following parameters (abbreviation in parentheses): chi-square, Goodness of Fit Index (GFI), Root Mean Square Error of Approximation (RMSEA), and Convergence Factor Index (CFI). According to the standards, the values of GFI, CFI > 0.9, and RMSEA < 0.05 suggest a good model fit [31].

## 3. Results

### 3.1. Adaptation of the ABAQ to Spanish (ABAQ-Mx)

The level of agreement among the experts (clarity and congruence of the items) was adequate. All 20 items obtained the maximum score for congruence (5 points). For clarity, 14 questions obtained the maximum score, while six items obtained values between three and four points (acceptable).

### 3.2. Psychometric Analysis of the ABAQ-Mx Questionnaire

A total of 261 health students participated in the study (82% female), with a mean age of 20.5 ± 2.5 years (Table 1).

The internal consistency of the ABAQ-Mx instrument with 20 items showed an α = 0.690. In the first step, four items (5, 7, 13, and 16) showed an item–total correlation ≤0.2 and were eliminated (Table 2). Then, the PCA was performed with the remaining 16 items. After confirming adequate sampling (KMO test; value 0.794), Bartlett’s test of sphericity (*p* ≤ 0.00) indicated relevant data to perform the PCA. We obtained factor loadings ≥0.3 on 13 items; the other three were eliminated for lower loadings. The items that were initially designed inversely maintained this structure.

According to the total variance percentage, the model is aligned to a single factor, explaining 24.42% of the questionnaire items’ variance. The ABAQ-13Mx was thus constituted with an α = 0.786, thus increasing the accuracy of the test. 

### 3.3. Replicability of Latent Structure (Confirmatory Factor Analysis)

To determine the adequacy and replicability of the ABAQ-13Mx’s factor structure in the Mexican population, the CFA was used. The results showed an adequate adjustment of the structure. With the 13 items, the model was statistically significant (*p* ≤ 0.001), with standardized factor loadings from 0.331 to 0.898. The statistical fit for this one-factor structural model without controlling for error was: chi-square = 163.585, gl:65, *p* < 0.001, CFI = 0.837, and RMSEA = 0.076. When controlling for errors, the fit indices improved significantly (chi-square = 98.413, df: 62, *p* = 0.02, CFI = 0.940, and RMSEA = 0.048). Figure 1 shows the factor structure of the ABAQ-13Mx items according to the CFA model.

## 4. Discussion

The ABAQ-13Mx was organized into a single factor valid for assessing BF attitudes in the participating Mexican higher education health students. Initially, according to the exploratory factor analysis, we obtained six factors with eigenvalues greater than unity that explained 57.4% of the total variance; however, according to the sedimentation plot, it was shown that the best solution was three factors that would explain 39.1% of the total variance of the items. However, after performing the confirmatory factor analysis, the factor weights of each subcomponents were not satisfactorily defined, suggesting that there are items that load on several factors or that perhaps there are items that do not correlate with the construct. Therefore, we adjusted the analysis to one factor, as well as a previous two studies done: both in the original ABAQ tool with 159 Australian participants [15] and in the adapted and validated ABAQ in 205 nursing students [20], they used a one-factor analysis, possibly for the same reason.

This questionnaire was reduced to 13 items (ABAQ-13Mx), as the seven questions eliminated did not significantly influence our sample to assess attitudes, not converging or not being specific enough for measuring the construct of BF attitudes.

Some of these eliminated questions dealt with aspects related to the mother. For example, “BF is a practice that increases the bond between mother and child” or “the mother loses great satisfaction of motherhood by not breastfeeding”. These items seem to have little relevance in our sample. Perhaps because the use of breast milk substitutes is normalized in Mexico, since the prevalence of BF is low (28.6%), or because of ignorance of the effects of breastfeeding on attachment. Due to the lack of social support for BF practice and the extensive and habitual recommendation of substitutes by health personnel themselves since the 1970s, the breastfeeding experience among mothers has been lost. So has the intergenerational transmission of how to breastfeed and the affective and health benefits it provides. Logically, feeding with milk substitutes can also establish an affective bond and provide satisfaction, a possible justification for why they could have been excluded. In the Chinese version of the ABAQ [20], the questions deleted related to the mothers were those asking smoking and alcohol drinking, which we kept in context.

Other elements eliminated (for example, “mothers should not breastfeed in public” or “a father feels displaced when a mother breastfeeds her baby”) do not seem to align with the construct of attitudes towards breastfeeding in context either. In Mexico, breastfeeding in public is not common; mothers who do so are often exposed to criticism, including harassment and various forms of gender discrimination [32]. Thus, Mexican breastfeeding mothers mentioned in a study that they preferred to breastfeed at home for convenience. The parents of infants even reported feeling uncomfortable when they saw mothers feeding their children in public places. The latter possibly derives from social conceptualizations built around the mammary glands, which are considered a sexual component rather than a source of nourishment [33]. It is necessary to modify the existing social norms about BF and strengthen the public policies supporting breastfeeding women to breastfeed without prejudice in public places [32].

Furthermore, there is still little male involvement in feeding, care, and child-rearing in Mexico [33]. Health professionals should provide adequate information on BF and seek the participation and integration of the father in this activity [34] and satisfy the needs of the mother [35], as these are critical factors for successful breastfeeding. It would be convenient to study these aspects in greater depth and a population with more male students. since our sample was primarily female. The Chinese validation [20] did not include, from the beginning, the original items 6 and 16, both related to mothers’ experiences related to BF. Brodribb et al. [15] maintained the 20 items, but, lately, both the Chinese [20] and this Mexican validation reduced this number to 12 and 13, respectively (the Chinese version do not include these items, according to the original numbering [15]: 3, 4, 6, 8, 11, 13, 16, and 17). Of the 12 items remaining in the Chinese version, only seven coincide with those remaining in the Mexican version, and only one item (item 13 in the original version) was eliminated for the same statistical criteria, low correlation.

Given that one of our main objectives was to have an instrument that would allow us to measure the phenomenon validly, we adjusted to meet the criteria required to perform the factor analysis adequately; that is, we considered 10 participants for each item [21]. The curricula of the bachelor’s degree program in nutrition and nursing have many identical subjects in basic common training (AFBC). Therefore, the basis of their training is similar, as we reviewed when analyzing the curricula of these degrees (information not shown). In medicine, the subjects tend to have a broader content and course duration. Additionally, the AFBC subjects are similar to those of the other two programs, which we assume allows us to bring these students together in the same study, as they are the most professionally focused on the mother–baby–dad triad. Besides, our sampling was focused on the internal validity of the tool; we did not seek the phenomenon’s representativeness in the entire population. On the other hand, we recognize that a limitation of our sample is that it does not allow us to generalize the results to all health students (psychology, dentistry, and sports, among others). Randomness was involved in selecting the groups (medicine, nursing, and nutrition) and determining the number of participants by randomly requesting access to the professors teaching in the classrooms we visited to apply the survey, according to the similarities in some aspects of the above-mentioned curricula. 

Another limitation is that the sample included only students from the thematic health center (UdeG) in an urban metropolitan area. In rural, regional centers, there are also these undergraduates; perhaps their training is different or may even include more/less knowledge and experience regarding breastfeeding. Thus, the study results could be influenced by the environment, the student’s educational trajectory, and are not representative of the totality of students in the Mexican health area. The items eliminated may not contribute to the concept of attitudes; however, they should be worked on socially, since they are indispensable elements for sustaining and strengthening the practice of BF. Maybe, maintaining the two sets of items (including the seven we eliminated) would allow us to analyze and present the results of the sub-part of the validated survey ABAQ-Mx 13 but also to see, if over time, in the case of longitudinal or cohort studies, we could see if other items, corresponding to attitudes intended to promote, through experiences and experiences, converge with the concept and contribute in creating a broader corpus of favorable attitudes and breastfeeding support.

As a strength of the present study, the ABAQ-13Mx tool is valid and reliable for evaluating BF attitudes of Spanish-speaking healthcare students. This tool can be applied in different university contexts. Specifically, in the UdeG network, there are several campuses with students of the same degrees (five campuses for medicine, eight for nutrition, and six for nursing); likewise, there are several educational institutions in the state of Jalisco that have the same study programs shared by the UdeG and verified by this same institution. It is a quick tool to apply and will allow creating educational strategies based on the results obtained. Before and after a specific educational intervention, it could also be used to improve these attitudes (course, breastfeeding counseling, and breastfeeding support groups) or even before and after clinical practices in a hospital or pre- and perinatal care contexts. 

With the use of this tool, we could also make precise diagnoses in specific contexts and, based on the responses obtained, specific educational strategies could be designed to improve the attitudes that in that context are not so favorable to the breastfeeding process.

Finally, working on attitudes towards breastfeeding during the training of students allows them to reflect, deepen and mature their attitudes. The latter would allow them to perform better in educational evaluations on attitudes regarding this topic, perhaps, but, above all, in the medium (during their training practice) and long term (as professionals), they could have a favorable impact thanks to those favorable attitudes.

## 5. Conclusions

The validated and reliable ABAQ-13 Mx is a tool to assess the attitudes towards breastfeeding among health students in our Spanish-speaking context. The use of this validated survey could be applied at different moments of the academic trajectory to progressively assess whether favorable changes are made in this corpus of attitudes. Therefore, this tool can be helpful to guide and reinforce the BF education training in health higher education Spanish-speaking students. That final impact of using this validated tool could help mothers, fathers, and family members provide more significant support to them and considerable benefits for children, mothers, and society through increased breastfeeding rates.

Perhaps its use can also be envisaged, although, in this case, we would suggest validating it again in health professionals who have already graduated or have established professional practice, a proposal that we plan to make at a later date.

## Figures and Tables

**Figure 1 ijerph-18-04609-f001:**
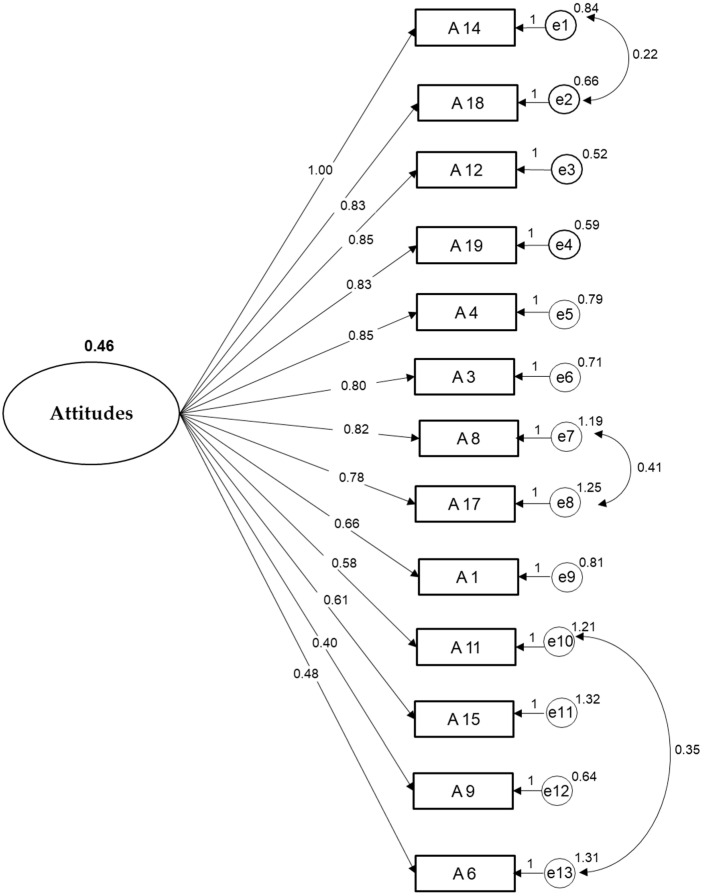
One-factor confirmatory analysis of the Spanish version of the ABAQ-13Mx.

**Table 1 ijerph-18-04609-t001:** Characteristics of the health sciences students who responded to the questionnaire.

Variables	Values	Total *n* (%)	Nutrition *n* (%)	Medicine *n* (%)	Nursing *n* (%)
Age	m (SD)	20.46 (2.4)	20.57 (3.5)	20.25 (1.3)	20.90 (2.1)
Sample		264 (100)	95 (36)	130 (49.2)	39 (14.8)
Sex					
	Women	214 (81)	84 (88.4)	97 (74.6)	33 (84.6)
	Men	50 (19)	11 (11.6)	33 (25.4)	6 (15.4)
Last completed semester					
	*2*	9 (3.4)	3 (3.2)	5 (3.8)	1 (2.6)
	*3*	92 (34.8)	21 (22.1)	59 (45.4)	12 (30.8)
	*4*	92 (34.8)	47 (49.5)	38 (29.2)	7 (17.9)
	*5*	43 (16.3)	19 (20.0)	16 (12.3)	8 (20.5)
	*6*	15 (5.7)	1 (1.1)	7 (5.4)	7 (17.9)
	*7*	13 (4.9)	4 (4.2)	5 (3.8)	4 (10.3)

**Table 2 ijerph-18-04609-t002:** Item–total correlations and exploratory factor analysis of the Spanish questionnaire ABAQ-13Mx (*n* = 264). The questions are presented both in English and Spanish.

Item	Question	Item-Total Correlations	Loading Factor
A 1 ^a^	Infant formula is more easily digested than human milk	0.485	0.499
	*La fórmula infantil se digiere más fácilmente que la leche humana*		
A 2	Human milk is the ideal food for babies	0.332	0.220 ^c^
	*La leche humana es el alimento ideal para los bebés*		
A 3 ^a^	Infant formula feeding is a good way to allow dad to participate in feeding for the baby	0.606	0.599
	*La alimentación fórmula infantil es una buena manera de dejar que el papá participe en la alimentación de bebé*		
A 4 ^a^	Infant formula is an acceptable method of feeding infants	0.483	0.576
	*La fórmula infantil es un método aceptado para alimentar a los bebés*		
A 5	Breastfeeding increases the mother-child bonding	0.180 ^b^	-
	*La lactancia aumenta la unión madre-hijo*		
A 6	A mother naturally and learns instinctively how to breastfeed her baby	0.301	0.369
	*Una madre aprende de manera natural e instintivamente cómo se debe amamantar a su bebé*		
A 7	Breastfeeding provides health benefits for babies that cannot be provided by infant formula	0.034 ^b^	-
	*La leche humana brinda beneficios en la salud de los bebés, que no son proporcionados en una fórmula infantil*		
A 8 ^a^	Mothers who smoke should infant formula feed their babies	0.506	0.538
	*Las madres que fuman deben alimentar a su bebé con fórmula infantil*		
A 9 ^a^	Breastfeeding is incompatible with the mother working outside the home	0.359	0.393
	*La lactancia materna es incompatible con el trabajo de la madre fuera del hogar*		
A 10 ^a^	Father feel left out if the mother breastfeeds her baby	0.221	0.195^c^
	*Un papá es desplazado cuando la madre amamanta a su bebé*		
A 11 ^a^	Breastfed babies need to be fed too often	0.427	0.438
	*Los bebés que son amamantados necesitan ser alimentados con mayor frecuencia*		
A 12 ^a^	Infant formula is as healthy for an infant as breast milk	0.592	0.633
	*La fórmula infantil es tan saludable para un bebé como la leche materna*		
A 13	Human milk is greater benefits than feeding infant formulas	0.046 ^b^	-
	*La leche humana tiene mayores beneficios que la alimentación con fórmulas infantiles*		
A 14 ^a^	Infant formula feeding is the best choice to feed your baby when the mother plans to go out to work	0.713	0.695
	*La alimentación con fórmula infantil es la mejor opción para alimentar a su bebé cuando la madre planea trabajar*		
A 15 ^a^	The benefits of breast milk last only when the baby is breastfed.	0.390	0.369
	*Los beneficios de la leche humana duran únicamente cuando el bebé es amamantado*		
A 16	Mothers who infant formula feed their baby miss one of the great joys/satisfactions that motherhood provides	0.119 ^b^	-
	*Las madres que alimentan a su bebé con fórmula infantil, se pierden una de las grandes alegrías/satisfacciones que proporciona la maternidad*		
A 17 ^a^	A mother who drinks alcohol should not breastfeed her baby	0.437	0.509
	*Una madre que consume alcohol no debería amamantar a su bebé*		
A 18 ^a^	Infant formula feeding is more reliable because the exact amount of milk the baby takes can be calculated	0.671	0.660
	*La alimentación con fórmula infantil es más confiable porque se puede calcular la cantidad exacta de leche que el bebé toma*		
A 19 ^a^	Current infant formulas are nutritionally equivalent compared to human milk	0.584	0.617
	*Las fórmulas infantiles actuales son nutricionalmente equivalentes en comparación con la leche humana*		
A 20 ^a^	Women should not breastfeed their babies in public places, such as restaurants	0.247	0.123 ^c^
	*Las mujeres no deberían amamantar a sus bebés en lugares públicos, como son los restaurantes*		

Note: ^a^. Reversely worded items. ^b^. Items with a Spearman correlation ≤ 0.2 were removed. ^c^. Items with a factor loading < 0.3 were also removed from the model.

## Data Availability

The database is available with authors.

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
