# Peer review of "Validation of the Spanish-Mexican Version of the Australian Breastfeeding Attitude Questionnaire in Higher Education Health Students"

_ijerph, 2021, doi:10.3390/ijerph18094609_

Round 1
Reviewer 1 Report
Introduction: Authors should elaborate why the study validated the BF measure in health students instead of the whole population. Although the authors have pointed out the strengths of ABKAQ in assessing BF knowledge and attitudes, it would be more helpful and interesting to compare it with other existing tools. This can better justify the use of ABKAQ in this study.
Methods: The authors mentioned that ten participants were considered for each item, but given that the sample consisted of students from three different health disciplines, and most were medical students, authors should comment how this may affect the results or how they have addressed this issue in the analyses.
Why was the translated tool evaluated by nutrition experts but not medicine or nursing experts when the study sample consisted of all three student groups (nutrition, medicine and nursing)?
Has the research protocol been reviewed and approved by an ethics committee?
How many medical schools in Mexico? What is the ranking of the medical school where recruitment took place in terms of student socioeconomic profile or academic achievement? This can give readers an idea of the extent to which the sample is representative of the Mexico health student population.
Results: Was the EFA conducted for one-factor solution only? The results would be more complete if higher number of factor solutions were also presented.
Conclusions: Authors stated that due to low relevance to their culture, seven items were eliminated. However, statistically speaking, they were eliminated because they showed little consistency with the other items in the scale. Authors should explain why those items must be eliminated to make a reliable scale, instead of focusing on their cultural relevance.
Authors mentioned Chinese validation has good psychometric properties. Is that the Chinese version also has the same 13 items? If not, can the authors comment why the Chinese version includes the item "mothers should not breastfeed in public" but the current Mexican version does not, given that the Chinese also has a conservative culture and is not common to breastfeed in public places.
The authors should elaborate more on the implications of the validated tool for practice and research
Reviewer 2 Report
The authors validated a Spanish-Mexican version of the Australian Breastfeeding Attitude Questionnaire. The field is very interesting and could be really useful for Spanish speakers. However I have doubts about the usefulness if its validated only in students, as the original version was designed to measure breastfeeding knowledge and attitudes of health professionals, not in students.
Moreover, the final version has deleted some items than, according to authors opinion, do not fit the mexican usual behaviour. In my opinion, this would be a good reason to mantain the two original dimension and give an accurate reflection of what really happens in Mexican society.
Round 2
Reviewer 1 Report
The authors have addressed all the comments.
Reviewer 2 Report
After the changes done, in my opinion, the article can be published in its present form.